# *Burkholderia* PglL enzymes are Serine preferring oligosaccharyltransferases which target conserved proteins across the *Burkholderia* genus

Andrew J. Hayes[1], Jessica M. Lewis[1], Mark R. Davies [1] & Nichollas E. Scott [1✉]

Glycosylation is increasingly recognised as a common protein modification within bacterial proteomes. While great strides have been made in identifying species that contain glycosylation systems, our understanding of the proteins and sites targeted by these systems is far more limited. Within this work we explore the conservation of glycoproteins and glycosylation sites across the pan-*Burkholderia* glycoproteome. Using a multi-protease glycoproteomic approach, we generate high-confidence glycoproteomes in two widely utilized *B. cenocepacia* strains, K56-2 and H111. This resource reveals glycosylation occurs exclusively at Serine residues and that glycoproteins/glycosylation sites are highly conserved across *B. cenocepacia* isolates. This preference for glycosylation at Serine residues is observed across at least 9 *Burkholderia* glycoproteomes, supporting that Serine is the dominant residue targeted by PglL-mediated glycosylation across the *Burkholderia* genus. Combined, this work demonstrates that PglL enzymes of the *Burkholderia* genus are Serine-preferring oligosaccharyltransferases that target conserved and shared protein substrates.

[1] Department of Microbiology and Immunology, University of Melbourne at the Peter Doherty Institute for Infection and Immunity, Melbourne, Australia.
✉email: Nichollas.scott@unimelb.edu.au

Glycosylation is a common class of protein modifications increasingly recognised within bacterial proteomes[1,2]. Within bacterial proteomes, the attachment of glycans has been shown to influence protein functions by enhancing protein stability[3–5], stabilising protein complexes[6], masking antigenic sites[7], and modulating protein enzymatic activities[8,9]. As these events fine-tune the proteome, identifying novel glycosylation systems has been a key goal of microbial glycoproteomics. Over the last decade, with the aid of mass spectrometry-based proteomics, a range of bacterial glycosylation systems have been identified with many of these now demonstrated to be conserved across genera[1,2,10,11]. Within these systems follow up studies have mostly focused on understanding the glycans used for protein glycosylation, defining their diversity[12–14] and elucidating their biosynthetic pathways[15–17]. This focus has resulted in a limited understanding of the specific glycoproteins and glycosylation sites within the majority of known bacterial glycoproteomes.

One of the best characterised bacterial glycoproteomes to date is that of the N-linked glycosylation system of *Campylobacter jejuni*, where 134 glycosylation sites have been experimentally identified[18]. The success in mapping glycosylation across the *C. jejuni* glycoproteome has largely been due to two features of this system; i) glycosylation events within the *Campylobacter* genus occurring within the glycosylation sequon, D/E-$X_1$-N-$X_2$-S/T (where neither $X_1$ or $X_2$ can be proline)[19], which restricts the possible glycosylation sites within a protein, and ii) the chemical nature of N-linked glycosylation that can allow localisation information to be obtained using collision-based Mass Spectrometry (MS) fragmentation approaches[20]. For bacterial mediated O-linked glycosylation systems, far less is known about the sites of glycosylation[1,2]. This limited understanding of bacterial O-linked glycosylation sites is largely driven by technical limitations associated with the analysis of O-linked glycopeptides, where the chemical nature of O-linked glycosylation requires approaches such as electron-transfer dissociation (ETD) or Electron-transfer/higher-energy collision dissociation (EThcD) to localise glycosylation sites[20]. Defining glycosylation sites are critical for enabling functional characterisation of glycoproteins, as well as for understanding the conservation of sites across species. From previous studies, it has become clear that bacterial glycosylation sites can vary even within closely related strains/species[21,22], and that understanding glycosylation site heterogeneity can provide critical insights into functionally important nuances within glycoproteins. An example of this can be seen in the pilin of *Neisseria* species, where high levels of glycosylation are correlated with masking of conserved antigenic regions, conversely, forms of pilin prone to amino acid sequence variation possess limited pilin glycosylation[7,17,23,24].

One of the most widespread families of O-linked glycosylating enzymes are the PglL oligosaccharyltransferases[25], which have been experimentally shown to O-glycosylate proteins within *Acinetobacter*[26,27], *Neisseria*[14,17,28], *Burkholderia*[13,15,29], *Francisella*[30], *Pseudomonas*[31], and *Ralstonia*[32] species. Within these species, PglL mediates O-linked glycosylation of substrates within the periplasmic space, and can be responsible for the modification of a single protein or multiple proteins depending on the enzyme[10,11]. To date, no rigid glycosylation sequon has been observed within PglL substrates, with glycosylation predominantly occurring in disordered regions rich in Alanine and Prolines[12,29,33,34]. For PglL enzymes responsible for the glycosylation of multiple proteins, known as general glycosylation systems, the abolishment of PglL has been shown to result in profound effects on virulence[27,29,32]. Despite this, the lack of a detailed understanding of most O-linked glycoproteomes has limited our ability to understand how glycosylation influences virulence as well as which glycosylation sites are critical for

virulence-associated processes. Understanding the proteins targeted for glycosylation thus could provide an important first step to improve our understanding of virulence within poorly understood pathogens such as *Burkholderia cenocepacia*.

The opportunistic pathogen *B. cenocepacia* is associated with life-threatening infections in people with cystic fibrosis (CF)[35]. While ubiquitous in the environment not all *B. cenocepacia* strains are associated with human disease with the phylogenetic grouping known as the IIIA genomovar overrepresented in CF infections[36–38]. Within the IIIA genomovar, the K56-2[39] and H111[40] strains have been established as the most widely used models of *B. cenocepacia* pathogenesis. While both strains were isolated from CF patients, only K56-2 is a member of the ET12 lineage, a highly transmissible lineage associated with high mortality rates[41,42]. The absence of key genetic elements in H111, such as the low-oxygen-activated locus (*lxa*)[43] and the cciIR quorum-sensing system[44,45], has been suggested to account for the notable differences in virulence traits[46] and quorum-sensing[47,48] observed between K56-2 and H111. This genetic diversity between IIIA genomovar isolates makes understanding multiple strains essential for identifying core features shared across the majority of pathogenic *B. cenocepacia* isolates.

Recently we began exploring the pan-glycoproteome of *Burkholderia* species focusing on the conservation of the glycans used for glycosylation[13,15] and the properties of glycopeptides that can influence their enrichment using zwitterionic hydrophilic interaction liquid chromatography (ZIC-HILIC)[49]. These studies have highlighted that the glycoproteomes of *Burkholderia* species appear larger than initially thought, yet, the lack of defined glycosylation sites associated with these proteins has limited our ability to understand the general trends within site utilisation across the *Burkholderia* pan-glycoproteome. To improve our understanding of *Burkholderia* glycosylation, we have undertaken a site focused glycoproteomic study of *Burkholderia cenocepacia* within K56-2 and H111. Leveraging this curated resource, we gain and experimentally confirm a previously unrecognised preference in the specificity of O-linked glycosylation, as well as demonstrate that *Burkholderia* glycosylation targets conserved protein substrates across this genus. Combined, this work demonstrates the high levels of O-linked glycosylation conservation across the *Burkholderia* genus.

## Results

**O-linked glycosylation targets similar proteins and glycosylation disruption impacts the proteome in a similar manner across *B. cenocepacia* strains.** Recently we noted that the physiochemical properties of peptides heavily influences the ability of ZIC-HILIC enrichment to isolate bacterial glycopeptides[49]. This observation suggests numerous glycoproteins have likely been overlooked when previous studies used Trypsin alone to assess the glycoproteome of *B. cenocepacia*[13,29]. To increase the coverage of the *B. cenocepacia* glycoproteome we undertook glycoproteomic analysis using multiple proteases[50,51] and two widely used *B. cenocepacia* strains, K56-2 and H111. Glycopeptide enrichments of Trypsin, Thermolysin, and Pepsin digested samples enabled the identification of 584 and 666 unique glycopeptides from K56-2 and H111 strains respectively (Fig. 1a, Supplementary Data 1 and 2). Although these glycopeptide datasets identified more glycoproteins than identified within previously published studies using the ET12 *B. cenocepacia* strain J2315 (Supplementary Fig. 1a[13,29]), the majority of identified glycoproteins were unique to a single strain (Supplementary Fig. 1b). This high degree of heterogeneity suggested the presence of erroneous assignments within the datasets, a known issue associated with the assignment of a relatively limited population

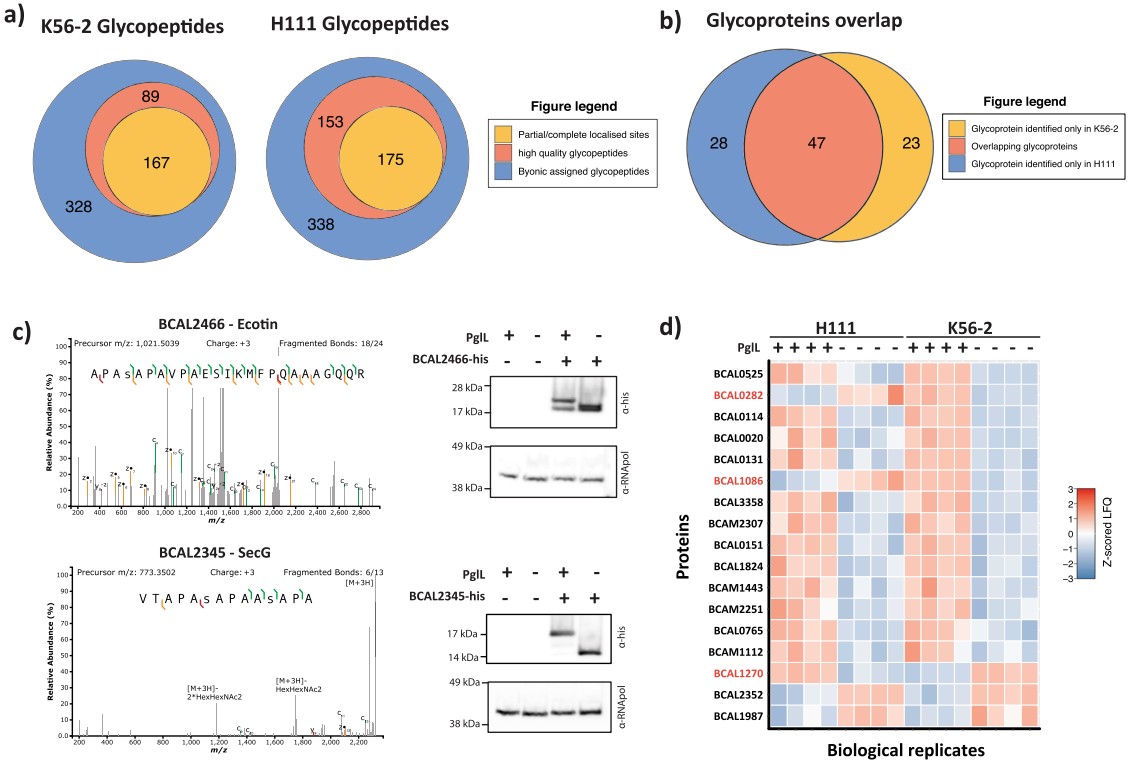

**Fig. 1 *B. cenocepacia* glycoproteomic analysis reveals similar substrates and functional impacts on *B. cenocepacia* strains. a** Within *B. cenocepacia* strains K56-2 and H111 584 and 666 unique glycopeptides were identified respectively (Byonic scores >300). Within these glycopeptides manual curation identified 256 and 328 glycopeptides as high-quality unique glycopeptides of which 167 and 175 provided partial or complete glycosylation site information. **b** Across the manually curated glycopeptides a total of 98 unique glycoproteins were identified of which 47 were identified within both strains. **c** Western blotting of two potential glycoproteins, His[6] tagged BCAL2466 and BCAL2345, reveals alterations in protein banding when expressed within the glycosylation null strain K56-2 Δ*pglL* compared to K56-2 WT with anti-RNA pol westerns included as loading controls. **d** LFQ analysis of the most altered proteins observed between H111 Δ*pglL* compared to H111 WT reveals similar proteome alterations as seen previously within K56-2 [5].

of modified peptides within large database searches[52]. Consistent with this hypothesis, examination of the glycopeptide scores reveal a bias toward lower scores (Supplementary Fig. 2a, b) supporting a higher than ideal false discovery rate despite stringent filtering[53]. To improve the quality of these glycopeptide datasets we manually curated the Byonic assigned glycopeptides, identifying 256 and 328 high-quality unique glycopeptides in K56-2 and H111 respectively (Fig. 1a; Supplementary Data 3). Of these unique glycopeptides, 167 and 175 glycopeptides within K56-2 and H111 respectively provided partial or complete site localisation (Fig. 1a; Supplementary Data 3). Consistent with improving the data quality, the curated glycopeptides revealed a Gaussian distribution within the scores, removing predominantly low scoring identifications (Supplementary Fig. 2c–f). Comparing the observed score distributions for each proteases revealed that for both H111 and K56-2 Thermolysin and Pepsin glycopeptides typically possessed lower and narrower score distributions, yet still enabled the localisation of glycosylation sites at a similar frequency as Trypsin (>50% of unique glycopeptides localised, Supplementary Fig. 3a–d). Of these manually assessed glycopeptides, we noted ~65% of glycoproteins were identified within both K56-2 and H111 strains (Fig. 1b), supporting a conserved glycoproteome across *B. cenocepacia* strains.

To validate the accuracy of these manually filtered assignments two small novel glycoproteins, BCAL2466 and BCAL2345, were His[6] epitope tagged and expressed within K56-2 WT and K56-2 Δ*pglL* strains (Fig. 1c). Consistent with a single glycosylation event on BCAL2466 and multiple glycosylation events on BCAL2345, the expression of these proteins within K56-2 Δ*pglL* resulted in increased gel mobility supporting their glycosylation

status. Within the wildtype strains, differences in the occupancy of these proteins were also noted, with no non-glycosylated protein observed for BCAL2345, contrasting BCAL2466 where both glycosylated and non-glycosylated forms were detected (Fig. 1c). Interestingly it should be noted that a subtle difference in the mobility of the loading control RpoA was also noted between K56-2 WT and K56-2 Δ*pglL* strains yet the cause of this is unknown (Fig. 1c). To functionally support the observed similarities within the glycoproteomes of *B. cenocepacia* strains, we generated two independent H111 Δ*pglL* mutants and assessed the proteome changes observed in the absence of glycosylation within H111, compared to our previously published K56-2 findings (Fig. 1d[5]). Although the proteomic alterations show congruent behaviour, it should be noted strain differences are observed within some proteins such as BCAL1086, BCAL0282, and BCAL1270 (Fig. 1d; Supplementary Data 4). Regardless of these differences between K56-2 and H111, the proteins observed to undergo statistically significant alterations within H111 Δ*pglL* were enriched for proteins observed to be altered within K56-2 Δ*pglL* (Fisher exact test p-value = 0.0006 Supplementary Data 5). Independent H111 Δ*pglL* mutants also showed similar alterations with the proteome and were highly similar (Fisher exact test p-value = $5.67 \times 10^{-15}$, Supplementary Data 4 and 5). Taken together these findings support that the O-linked glycoproteome of *B. cenocepacia* strains are similar in substrates and functional consequences.

**Glycosylation-site analysis reveals that *B. cenocepacia* glycosylation occurs solely on Serine residues.** Our manually curated

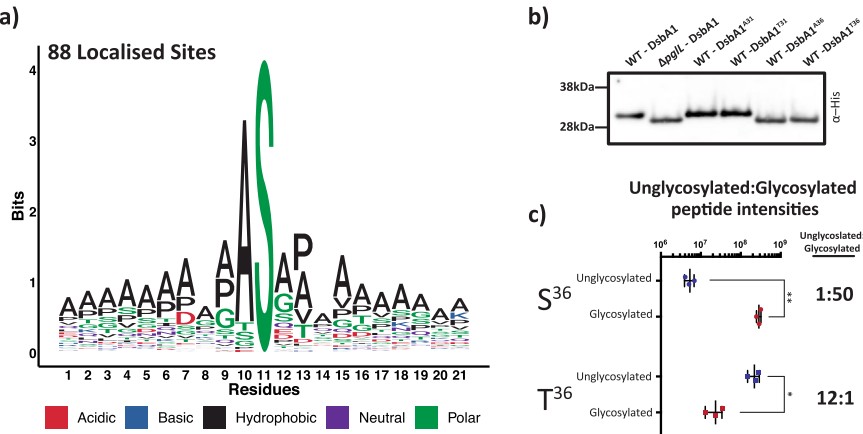

**Fig. 2 O-linked glycosylation predominantly occurs on Serine residues across the *B. cenocepacia* glycoproteome. a** Sequence analysis of localised glycosylation sites across *B. cenocepacia* strains reveals all assigned sites are Serine. **b** Western analysis of DsbA1$_{Nm}$-his$_6$ variants reveals substitution of only S$^{36}$ to Alanine or Threonine results in reduced glycosylation. **c** Relative amounts of the DsbA1$_{Nm}$-his$_6$ glycosylated and non-glycosylated peptides $^{23}$VQTSVPADSAPAASAAAAPAGLVEGQNYTVLANPIPQQQAGK$^{64}$ and $^{23}$VQTSVPADSAPAATAAAAPAGLVEGQNYTVLANPIPQQQAGK$^{64}$ observed within K56-2 WT reveals the alteration of S$^{36}$ to T$^{36}$ dramatically reduces glycosylation. Glycosylation site within peptides denoted by underlining. Differences in the abundance of glycosylated and non-glycosylated peptides containing S$^{36}$ to T$^{36}$ correspond to ($N = 3$) biologically independent samples assessed using two-sided t-tests resulting in p-values of 0.0054 and 0.036 respectively.

glycoproteomic data revealed >50% of unique glycopeptides provided at least partial site localisation information (Fig. 1a, Supplementary Fig. 3, Supplementary Data 4) of which 88 sites could be precisely localised across 70 glycoproteins (Supplementary Data 6). Surprisingly this analysis suggested a single Threonine residue, T$^{159}$ within BCAM0996, was modified within H111. Due to the discordance of this assignment with respect to all other sites (Supplementary Fig. 4), we sought to confirm the accuracy of BCAM0996 T$^{159}$. Examination of glycopeptides from BCAM0996 revealed that multiple Serine residues are modified within the same peptide observed to be modified at T$^{159}$ (Supplementary Data 3). The close proximity of this sole Threonine modification to multiple Serine modification events further raised concerns of miss-localisation. Manual annotation of the assigned glycopeptide supported the incorrect localisation of the glycosylation site, due to the incorrect assignment of the glycan and a secondary modification within the peptide sequence, ultimately revealing S$^{167}$ to be the correct localisation site (Supplementary Fig. 5). This finding suggested all localised glycosylation sites within both *B. cenocepacia* strains were observed on Serine residues. Examination of these 88 sites demonstrates that glycosylation favoured Alanine at the -1 position, yet this was not a strict requirement (Fig. 2a). To independently validate this preference for Serine, we re-analysed these glycoproteomic datasets with O-Pair, a glycosylation site localisation focused glycoproteomic tool[54], supporting that the majority of high-quality localisable glycosylation events occurred on Serine residues (Supplementary Fig. 6; Supplementary Data 7). Taken together, these findings support that glycosylation within the *B. cenocepacia* glycoproteome localises exclusively to Serine residues.

**Threonine residues undergo poor glycosylation in *B. cenocepacia*.** To experimentally explore this preference for Serine, we utilised the glycoprotein DsbA1$_{Nm}$-his$_6$ that we have previously observed is predominantly glycosylated by a single glycosylation event when expressed in *B. cenocepacia*[5,29]. Past work showed that glycosylation occurs at an unknown site within the peptide $^{23}$VQTSVPADSAPAASAAAAPAGLVEGQNYTVLANPIPQQQAGK$^{64}$, suggesting that one of the three Serine residues (S$^{26}$, S$^{31}$ and S$^{36}$) was the site of glycosylation within DsbA1$_{Nm}$-his$_6$. Of these sites, S$^{31}$ and S$^{36}$ are flanked by Alanine residues, consistent

with the preferred glycosylation sites within native substrates (Fig. 2a), yet only substitution of S$^{36}$ with either Alanine or Threonine resulted in gel mobility shifts consistent with the loss of glycosylation (Fig. 2b). Curiously DsbA1$_{Nm}$-his$_6$ A$^{36}$ leads to a slight but reproducible increased mobility compared to other point mutants yet our proteomic and sequencing result support the correctness of this construct, and as such the cause of this shift remains unknown. Glycopeptide analysis further confirmed S$^{36}$ was the sole residue modified within this peptide (Supplementary Fig. 7 and Supplementary Data 8). As MS analysis provides greater dynamic range then western blotting, we investigated if glycosylation still occurred at T$^{36}$ with analysis supporting the presence of a glycosylated form of the $^{23}$VQTSVPAD SAPAATAAAAPAGLVEGQNYTVLANPIPQQQAGK$^{64}$ at <10% of the abundance of the unmodified form (Fig. 2c, Supplementary Fig. 8a, Student's *t* test p-values = 0.036). Targeted MS analysis supported the identity of this glycopeptide, yet could not provide definitive site localisation to T$^{36}$ (Supplementary Fig. 8b). To further assess if Threonine could be modified, we introduced the DsbA1$_{Nm}$-his$_6$ point mutants T$^{36}$ and A$^{36}$ into K56-2 Δ*pglL amrAB::S7-pglL-his$_{10}$*, an overexpressing PglL$_{BC}$ strain[5], to assess if increasing PglL$_{BC}$ levels enhanced Threonine glycosylation. Western analysis demonstrates the majority of DsbA1$_{Nm}$-his$_6$ T$^{36}$ remained unglycosylated within K56-2 Δ*pglL amrAB::S7-pglL-his$_{10}$*. Yet, in contrast to K56-2 WT an additional faint band is observable within DsbA1$_{Nm}$-his$_6$ T$^{36}$ (Fig. 3a), consistent with the presence of low abundance glycosylated DsbA1$_{Nm}$-his$_6$ T$^{36}$. Targeted MS analysis confirms the glycosylation of T$^{36}$ in DsbA1$_{Nm}$-his$_6$ T$^{36}$ when expressed in K56-2 Δ*pglL amrAB::S7-pglL-his$_{10}$* (Fig. 3b, c). Combined this data supports that it is possible to glycosylate Threonine in *B. cenocepacia*, but the preferred residue for glycosylation is Serine, even when PglL$_{BC}$ is overexpressed.

**Glycoproteins and glycosylation sites are highly conserved across *B. cenocepacia* strains.** With our proteome and site-directed analysis supporting Serine as the favoured residue for glycosylation, we next addressed whether these glycosylation proteins/sites are conserved across a diverse set of *B. cenocepacia* strains. Leveraging our curated glycoprotein dataset (Supplementary Data 6) we assessed the conservation of these 70

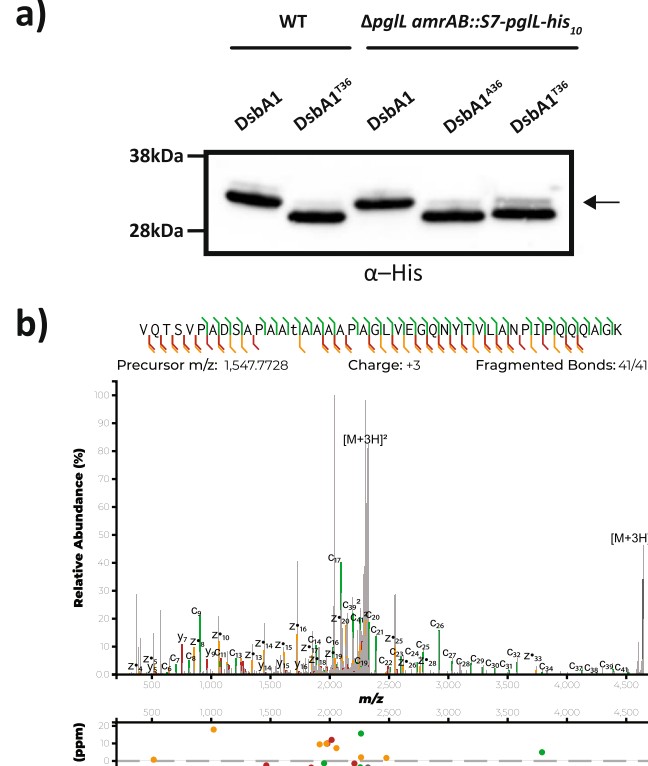

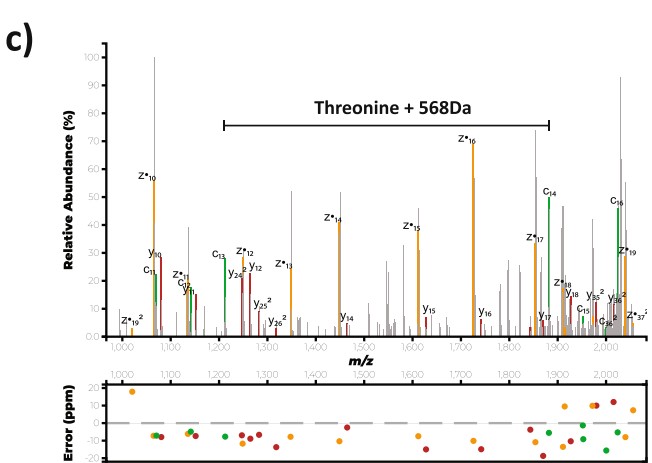

**Fig. 3 Threonine residues undergo poor glycosylation in _B. cenocepacia_ even with overexpression of _pglL_BC. a** Western analysis of DsbA1$_{Nm}$-his$_6$ variants reveals a minor product, indicated with an arrow, is observed within DsbA1$_{Nm}$-his$_6$ T$^{36}$ when expressed within K56-2 Δ_pglL amrAB::S7-pglL-his_10. **b**, **c** Targeted MS analysis using EThcD fragmentation enabled the confirmation of glycosylation at residue T$^{36}$ within $^{23}$VQTSVPADSAP AA_T_AAAAPAGLVEGQNYTVLANPIPQQQAGK$^{64}$. Glycosylation site within peptide denoted by underlining.

glycoproteins across 294 publicly available _B. cenocepacia_ genomes[55]. Across strains, the frequency of observation, defined as the presence of proteins with a minimum blast identity of >80%, revealed 65 glycoproteins are present in >80% of available genomes and that these glycoproteins are highly similar, with an average amino acid sequence identity >95% (Fig. 4a, Supplementary Data 9). This high level of conservation is also reflected at the glycosylation site (±10 amino acids) where the majority of glycosylation sites are highly conserved at the sequence level (Fig. 4b, Supplementary Data 9). It should be noted that although

the majority of glycosylation sites are conserved (Supplementary Figs. 9–14), variations are observed; such as in S$^{140}$ of BCAL1746, which is present in only a fraction of strains (Fig. 4c); S$^{404}$ of BCAL1674, where multiple alterations within the flanking sequences were noted (Fig. 4e); and S$^{262}$ of BCAL0163, where the serine required for glycosylation is lost across multiple strains (Fig. 4f). These data support that both _B. cenocepacia_ glycoproteins and glycosylation sites are highly conserved across _B. cenocepacia_ strains.

**Across _Burkholderia_ species similar glycoproteins are targeted for glycosylation at Serine residues.** In light of the high sequence identity of PglL across _Burkholderia_ species (Supplementary Fig. 15) we sought to assess if the preference for Serine glycosylation was a common feature across _Burkholderia_ glycoproteomes. To investigate this, we utilised our recently published glycoproteomic datasets of eight _Burkholderia_ species (_B. pseudomallei_ K96243; _B. multivorans_ MSMB2008; _B. dolosa_ AU0158; _B. humptydooensis_ MSMB43; _B. ubonensis_ MSMB22; _B. anthina_ MSMB649; _B. diffusa_ MSMB375 and _B. pseudomultivorans_ MSMB2199[13]) and re-analysed these datasets using O-Pair. As with _B. cenocepacia_, Serine was the dominant residue targeted for glycosylation with only 11 out of 440 high-confidence sites localised to Threonine (Fig. 5a; Supplementary Data 10). To experimentally support this preference, we introduced DsbA1$_{Nm}$-his$_6$ variants into two strains, _B. humptydooensis_ MSMB43 and _B. ubonensis_ MSMB22, to compare the preference for glycosylation at Serine over Threonine residues. Similar to _B. cenocepacia_, we find DsbA1$_{Nm}$-his$_6$ T$^{36}$ is predominantly unable to be glycosylated within both _B. humptydooensis_ MSMB43 and _B. ubonensis_ MSMB22 (Fig. 5b, c). As with _B. cenocepacia_, the glycosylated form of the peptide $^{23}$VQTSVPADSAPAA_T_AAAAPAGLVEG QNYTVLANPIPQQQAGK$^{64}$ was observed at low levels within both _B. humptydooensis_ MSMB43 and _B. ubonensis_ MSMB22 (Supplementary Fig. 16; Supplementary Data 11). Finally, to survey the similarities between _Burkholderia_ glycoproteomes we assessed the sequence identity of glycoproteins revealing the majority of glycoproteins are homologues of confirmed glycoproteins within _B. cenocepacia_ (Fig. 5d, Supplementary Data 12). Enrichment analysis supports that for _B. anthina_ MSMB649; _B. ubonensis_ MSMB22; _B. multivorans_ MSMB2008; _B. pseudomultivorans_ MSMB2199, and _B. dolosa_ AU0158, this overlap represents a statistically significant enrichment (Fig. 5e Supplementary Data 12). Combined these results support that Serine is the preferred target of glycosylation across _Burkholderia_ glycoproteomes and that similar glycoproteins are targeted across the _Burkholderia_ genus.

## Discussion

The conservation of glycosylation systems across bacterial species is increasingly being recognised as a common phenomenon in bacterial genera[14,56,57]. Although previous studies have sought to confirm the presence of protein glycosylation as well as highlight differences within the glycans utilised across species, little attention has been given to the protein's substrates and glycosylation sites themselves. Within this work we undertook a glycosylation-site-focused analysis of _Burkholderia_ species, revealing both experimentally (Figs. 1b, 5d) and bioinformatically (Fig. 4a) that proteins subjected to glycosylation are conserved across _Burkholderia_ strains/species and that O-linked glycosylation within this genus is overwhelmingly restricted to Serine residues (Figs. 2a, 5a). Our findings here highlight that protein glycosylation targets nearly 100 protein substrates, the majority of which appear conserved and targeted for glycosylation across multiple _Burkholderia_ species, supporting that these glycosylation events

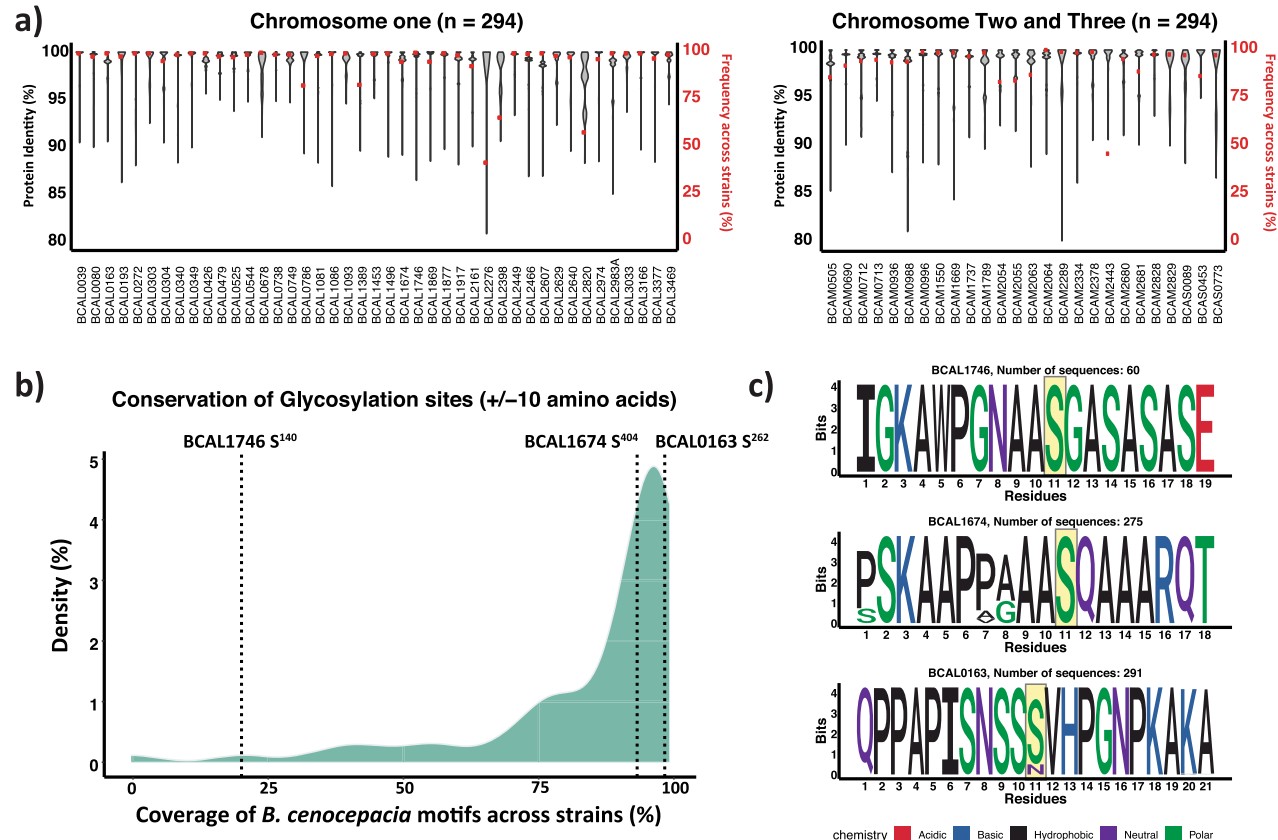

**Fig. 4 Glycoproteins and glycosylation sites are conserved across _B. cenocepacia_. a** Violin plot showing the amino acid sequence variation and frequency of 69 glycoproteins across 294 _B. cenocepacia_ genomes. Relative frequency is shown in red (right axis) and protein sequence identity (>80% minimum) is shown in black (violin plot, left axis). **b** Density plot representing the conservation of the glycosylation site sequence across 69 glycoproteins within 294 _B. cenocepacia_ genomes **c** Sequence logos of glycoproteins denoted within the density plot; BCAL1746 S[140]; BCAL1674 S[404] and BCAL0163 S[262].

are responsible for shared, yet still largely unknown, functions across the _Burkholderia_ genus. Consistent with this concept, phenotypic studies have previously shown the loss of _O_-linked glycosylation leads to similar impacts on _Burkholderia_ species, such as in the case of glycosylation null _B. cenocepacia_, _B. pseudomallei_, and _B. thailandensis_ strains, which all possess biofilm defects[5,58]. Thus, taken together this highlights that _Burkholderia_ glycoproteomes are far more similar and conserved then previously noted.

Similar to the optimal motif proposed for the _N. meningitidis_ PglL, WPAAASAG[59], our proteomic analysis supports that Serine residues flanked by Alanine are predominantly targeted for glycosylation within _B. cenocepacia_, yet glycosylation can still occur within divergent sequences lacking consensus residues (Fig. 2a). An unexpected finding within this study is that Serine is strongly targeted for glycosylation across _Burkholderia_ glycoproteomes, and that a dramatic difference in the extent by which Threonine/Serine residues can be glycosylated exists. In fact, although multiple studies have examined the glycan promiscuity and general ability of PglL enzymes to glycosylate proteins[60–62], no previous reports have suggested that Threonine/Serine residues are not equivalent targets for glycosylation. This difference in preference between these residues is striking with semi-quantitative comparisons of the unglycosylated and glycosylated peptides suggesting Serine is glycosylated ~500 times more efficiently than Threonine within _B. cenocepacia_ (Fig. 2b, c), with a similar trend also noted within _B. humptydooensis_ MSMB43 and _B. ubonensis_ MSMB22 (Supplementary Fig. 16). Although we confirmed that Threonine can be glycosylated at a low occupancy, this dramatic difference in preference supports that Threonine

glycosylation events are likely rare in _Burkholderia_ glycoproteomes, or if observed, are likely found at modest levels of occupancy. Interestingly, within other PglL glycosylation systems where only a limited number of glycosylation sites have been confirmed to date, such as in _N. gonorrhoeae_[33,34], _A. baumannii_[12] and _F. tularensis_[30], all confirmed glycosylation sites appear to occur on Serine residues. These trends suggest that Serine specificity may be a more general feature of the PglL oligosaccharyltransferases, yet further work is required to confirm this preference across known PglL enzymes. Although Serines are the preferred glycosylated residue across _Burkholderia_ glycoproteins, it is unclear if all glycosylated Serine's are modified to a high level of occupancy. Our western blot analysis of His[6] tagged BCAL2466 and BCAL2345 (Fig. 1c) supports that differences in glycosylation occupancy do exist, yet whether specific sequence or protein characteristics predict this efficiency are unclear. Together, this suggests further studies are required to understand the properties which promote occupancy at specific sites.

At the genus, level our analysis supports that across _Burkholderia_ species glycoproteins are conserved (Fig. 5d and Supplementary Fig. 17). Although the limited number of known glycosylation sites within other bacterial glycosylation systems has hampered the analysis of glycosylation site conservation, the identification of varying numbers of glycosylation sites in proteins such as pilin of _Neisseria_ species[7,17,23,24] has suggested that glycosylation sites may be highly variable across strains/species. In contrast, our analysis highlights that the majority of glycosylation sites, within _B. cenocepacia_ at least, appear conserved and that loss of glycosylation sites appear rare. This said, we did note examples of glycosylation sites which were only observed within a

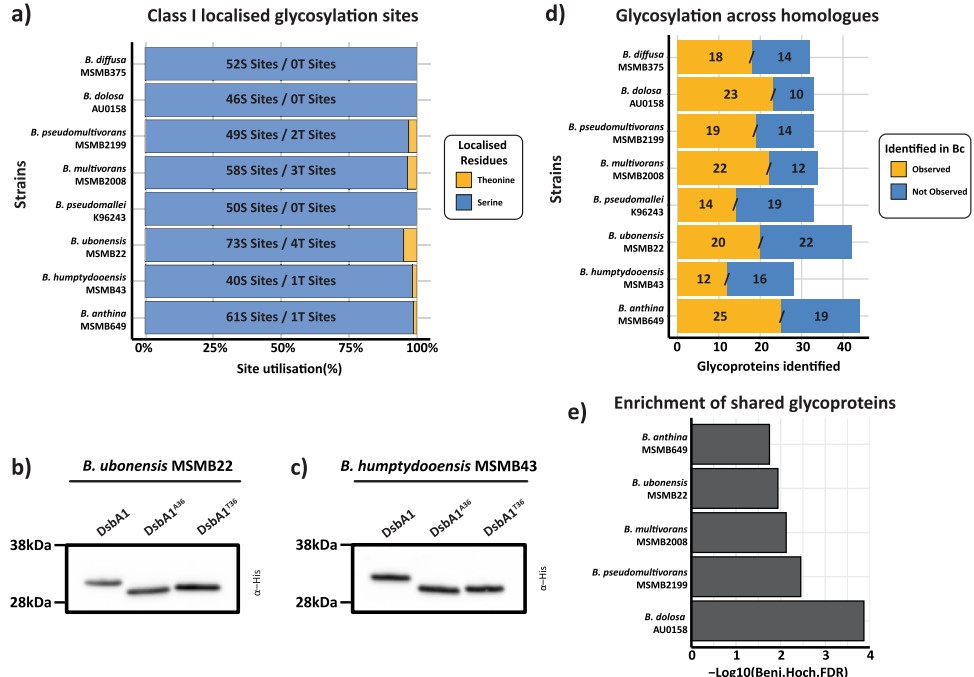

**Fig. 5 Serine glycosylation and the conservation of glycoproteins across *Burkholderia* species. a** The re-analysis of *Burkholderia* glycopeptide enrichment datasets reveals the majority of assigned glycosylation sites are localised to Serine residues. **b, c** Western analysis of DsbA1$_{Nm}$-his$_6$ variants expressed within *B. ubonensis* MSMB22 (**b**) and *B. humptydooensis* MSMB43 (**c**) supports that Threonine is disfavoured for glycosylation similar to *B. cenocepacia*. **d** Analysis of glycoproteins observed across *Burkholderia* species supports multiple *B. cenocepacia* glycoprotein homologues are observed glycosylated within other species. **e** Enrichment analysis of glycoproteins shows an enrichment of shared glycoproteins in *B. anthina* MSMB649; *B. ubonensis* MSMB22; *B. multivorans* MSMB2008; *B. pseudomultivorans* MSMB2199 and *B. dolosa* AU0158 compared to *B. cenocepacia*.

subset of genomes, such as S$^{140}$ of BCAL1746, yet this level of variability was the exception, not the rule, across *B. cenocepacia*. Although highly conserved, it is important to note at the glycoproteome level we observed only 47 of the 98 glycoproteins identified within both strains (Fig. 1b). Examination of the proteomes of H111 and K56-2 supports that this modest overlap is not due to the lack of expression of these glycoproteins, as at least 66 of these glycoproteins are expressed in both strains (Supplementary Fig. 18). Rather, this overlap supports that the differences in the observable glycoproteome are driven by either under-sampling of the glycoproteome due to the low abundance of these glycoproteins, or differences in glycosylation occupancy between strains. Nonetheless our glycoproteome analysis of *B. cenocepacia* significantly expands the known number of glycosylation sites from six sites in 23 glycoproteins[29] to 88 sites in 70 glycoproteins. It is important to note these glycosylation events likely still only represent a subset of the complete *B. cenocepacia* glycoproteome, as it is well known that growth under different conditions, such as nutrient limitation[63] and or in the presence of chemical queues[64,65] can dramatically alter the observable proteome of *B. cenocepacia*. As the growth of *B. cenocepacia* strains were undertaken under laboratory conditions with growth on rich media, it is possible multiple glycoproteins not expressed under these conditions have been missed. Thus, to better understand glycoproteins associated with virulence or survival under specific environmental conditions further studies may be required.

In summary, this work furthers our understanding of the breadth of the *B. cenocepacia* glycoproteome and the general features of glycosylation across members of the *Burkholderia* genus. The identification that the *B. cenocepacia* glycoproteome is far larger than initially thought, containing at least 98 proteins between strains, supports that glycosylation plays a multifaceted and pleiotropic role within *Burkholderia* species. The curation of a high-quality list of known glycoproteins and sites provides a

unique resource to facilitate studies that work towards understanding the roles of glycosylation within glycoproteins. From a mechanistic and technical standpoint, the insights into the preference for glycosylation at Serine residues improves our ability to predict the specific sites likely to be modified in glycoproteins, as well as improves our ability to assign glycosylation within proteomic datasets. Finally, the demonstration that glycoproteins are highly conserved across *Burkholderia* species also provides a new opportunity to use comparative glycoproteomics to dissect the conserved roles of glycoproteins across this genus. Combined these insights will aid in future studies to understand why glycosylation events are common and have been maintained across *Burkholderia* species.

## Methods

**Bacterial strains and growth conditions.** Strains and plasmids used in this study are listed in Supplementary Tables 1 and 2, respectively. Strains of *Escherichia coli* and *B. cenocepacia* were grown at 37 °C on Lysogeny Broth (LB) medium. When required, antibiotics were added to a final concentration of 50 μg/ml trimethoprim for *E. coli* and 100 μg/ml for *B. cenocepacia*, 20 μg/ml tetracycline for *E. coli* and 150 μg/ml for *B. cenocepacia* and 40 μg/ml kanamycin for *E. coli*. Ampicillin was used at 100 μg/ml and polymyxin B at 25 μg/ml for triparental mating to select against donor and helper *E. coli* strains as previously described[66]. Induction of tagged glycoproteins within *Burkholderia* strains was undertaken by the addition of either Rhamnose (final concentration 0.1%, for pSCrhaB2-based plasmids) or Arabinose (final concentration 0.5%, for pKM4-based plasmids) to overnight cultures. Antibiotics were purchased from Thermo Fisher Scientific while all other chemicals, unless otherwise stated, were provided by Sigma-Aldrich.

**Recombinant DNA methods.** Oligonucleotides used in this study are provided in Supplementary Table 3. The inducible pSCrhaB2-BCAL2345-his$_6$ and pSCrhaB2-BCAL2466-his$_6$ constructs were generated using Gibson assembly[67] by inserting PCR amplified fragments into *Nde*I and *Xba*I linearised pSCrhaB2 using NEBuilder® HiFi DNA master mix according to the manufacturer's instructions (New England Biolabs). pKM4 Site-directed mutagenesis was undertaken using PCR-based site replacement and *Dpn*I digestion[68]. All restriction endonuclease digestions, and agarose gel electrophoresis were performed using standard molecular

biology techniques[68]. All restriction enzymes were used according to the manufacturer's instructions (New England Biolabs). Chemically competent *E. coli* pir2 cells were transformed using heat shock-based transformation[68]. PCR amplifications were carried out using Phusion DNA polymerase (Thermo Fisher Scientific) according to the manufacturer's recommendations with the addition of 2.5% DMSO for the amplification of *B. cenocepacia* DNA, due to its high GC content. Genomic DNA isolations were performed using genomic DNA clean-up Kits (Zmyo research), while PCR recoveries and restriction digest purifications were performed using DNA Clean & Concentrator Kits (Zmyo research). Colony and screening PCRs were performed using GoTaq DNA polymerase (Promega; supplemented with 10% DMSO when screening *B. cenocepacia* gDNA). DNA sequencing was undertaken at the Australian Genome Research Facility (Melbourne, Australia).

**Construction of unmarked H111 Δ*pglL* deletion mutants**. Deletions of H111 *pglL* (Gene accession: I35_RS13570) were undertaken using the approach of Flannagan et al. for the construction of unmarked, non-polar deletions in *B. cenocepacia*[69] using the plasmid pYM8[15].

**Protein Immunoblotting**. Bacterial whole-cell lysates were prepared from overnight LB cultures of *Burkholderia* strains. 1 ml of overnight cultures at an OD$_{600}$ of 1.0 were pelleted, resuspended in 1X Laemmli loading buffer [24.8 mM Tris, 10 mM glycerol, 0.5% (w/v) SDS, 3.6 mM β-mercaptoethanol and 0.001% (w/v) of bromophenol blue (pH 6.8)] and heated for 10 min at 95 °C. Lysates were then subjected to SDS-PAGE using pre-cast 4-12% gels (Invitrogen) and transferred to nitrocellulose membranes. Membranes were blocked for 1 h in 5% skim milk in TBS-T (20 mM Tris, 150 mM NaCl and 0.1% Tween 20) and then incubated for at least 16 h at 4 °C with either mouse monoclonal anti-His (1:2,000; AD1.1.10, Biorad) or mouse anti-RNA pol (1:5,000; 4RA2, Neoclone). Proteins were detected using anti-mouse IgG horseradish peroxidase (HRP)-conjugated secondary antibodies (1:3,000; catalog number NEF822001EA, Perkin-Elmer) and developed with Clarity Western ECL Substrates (BioRad). All antibodies were diluted in TBS-T with 1% bovine serum albumin (BSA; Sigma-Aldrich). Images were obtained using an Amersham imager 600 (GE life sciences) or a Biorad ChemiDoc imaging station (Biorad).

**Preparation of cell lysates for proteomic analysis**. *B. cenocepacia* strains were grown overnight on LB plates as previously described[5]. Plates were flooded with 5 ml of pre-chilled sterile phosphate-buffered saline (PBS) and cells collected with a cell scraper. Cells were washed 3 times in PBS and collected by centrifugation at 10,000 × g at 4 °C then snap frozen. Frozen whole-cell samples were resuspended in 4% SDS, 100 mM Tris pH 8.0, 20 mM Dithiothreitol (DTT) and boiled at 95 °C with shaking for 10 min. Samples were then clarified by centrifugation at 17,000 × g for 10 min, the supernatant collected, and protein concentration determined by bicinchoninic acid assays (Thermo Fisher Scientific). For glycoproteomic analysis, 1 mg of protein for each sample (three biological replicates per strain/per enzyme) was acetone precipitated by mixing 4 volumes of ice-cold acetone with one volume of sample. For quantitative proteomic comparisons of H111 strains, 200 µg of protein for each biological replicate (four biological replicates per strain) were acetone precipitated by mixing 4 volumes of ice-cold acetone with one volume of sample. Samples were precipitated overnight at −20 °C and then centrifuged at 10,000 × g for 10 min at 0 °C. The precipitated protein pellets were resuspended in 80% ice-cold acetone and precipitated again for an additional 4 h at −20 °C. Following incubation, samples were spun down at 17,000 × g for 10 min at 0 °C to pellet precipitated protein, the supernatant discarded, and excess acetone evaporated at 65 °C for 5 min.

**Digestion of proteome samples for glycoproteomic analysis**. Dried protein pellets were resuspended in 6 M urea, 2 M thiourea, 50 mM NH$_4$HCO$_3$ and reduced with 20 mM DTT for 1 h followed by alkylation with 40 mM Iodoacetamide in the dark for 1 h[70]. Samples were then digested with one of three different protease combinations; (1) Trypsin (Promega) and Lys-c (Wako); (2) Thermolysin (Promega) or (3) Pepsin (Promega). (1) For the Trypsin/Lys-C digests; Lys-C (1/200 w/w) was added to reduced/alkylated samples for 4 h at room temperature before the sample was diluted with 100 mM NH$_4$HCO$_3$ four-fold to reduce the urea/thiourea concentration below 2 M and trypsin (1/50 w/w) added. (2) For Thermolysin digestions reduced/alkylated samples were diluted with 100 mM NH$_4$HCO$_3$ four-fold to reduce the urea/thiourea concentration below 2 M and Thermolysin (1/25 w/w) added. (3) For Pepsin digests reduced/alkylated samples were diluted with 0.1% TFA four-fold to reduce the urea/thiourea concentration below 2 M, the pH was confirmed to be pH ~2 and Pepsin (1/25 w/w) added. Digests were allowed to proceed overnight at room temperature with shaking. Digested samples were acidified to a final concentration of 0.5% formic acid and desalted on 50 mg tC18 Sep-Pak columns (Waters corporation) according to the manufacturer's instructions. tC18 Sep-Pak columns were conditioned with 10 bed volumes of Buffer B (0.1% formic acid, 80% acetonitrile), then equilibrated with 10 bed volumes of Buffer A* (0.1% TFA, 2% acetonitrile) before use. Samples were loaded on to equilibrated columns then columns washed with at least 10 bed volumes of Buffer A* before bound peptides were eluted with Buffer B. Eluted

peptides were dried by vacuum centrifugation at room temperature and stored at −20 °C.

**ZIC-HILIC enrichment of glycopeptides**. ZIC-HILIC enrichments were performed as according to the protocol of Mysling et al. with minor modifications[71]. ZIC-HILIC Stage tips[72] were created by packing 0.5 cm of 10 µm ZIC-HILIC resin (Millipore/Sigma) into p200 tips containing a frit of C8 Empore™ (Sigma) material. Prior to use, the columns were washed with Milli-Q water, followed by 95% acetonitrile and then equilibrated with 80% acetonitrile and 1% TFA. Digested proteome samples were resuspended in 80% acetonitrile and 1% TFA. Samples were adjusted to a concentration of 5 µg/µL (a total of 500 µg of peptide used for each enrichment) then loaded onto equilibrated ZIC-HILIC columns. ZIC-HILIC columns were washed with 20 bed volumes of 80% acetonitrile, 1% TFA to remove non-glycosylated peptides and bound peptides eluted with 10 bed volumes of Milli-Q water. Eluted peptides were dried by vacuum centrifugation at room temperature and stored at −20 °C.

**LC-MS analysis of glycopeptide enrichments**. ZIC-HILIC enriched samples were resuspended in Buffer A* and separated using a two-column chromatography set up composed of a PepMap100 C18 20 mm×75 µm trap and a PepMap C18 500 mm×75 µm analytical column (Thermo Fisher Scientific). Samples were concentrated onto the trap column at 5 µL/min for 5 min with Buffer A (0.1% formic acid, 2% DMSO) and then infused into an Orbitrap Fusion™ Lumos™ Tribrid™ Mass Spectrometer (Thermo Fisher Scientific) at 300 nl/min via the analytical column using a Dionex Ultimate 3000 UPLC (Thermo Fisher Scientific). 185-min analytical runs were undertaken by altering the buffer composition from 2% Buffer B (0.1% formic acid, 77.9% acetonitrile, 2% DMSO) to 28% B over 150 min, then from 28% B to 40% B over 10 min, then from 40% B to 100% B over 2 min. The composition was held at 100% B for 3 min, and then dropped to 2% B over 5 min before being held at 2% B for another 15 min. The Lumos™ Mass Spectrometer was operated in a data-dependent mode automatically switching between the acquisition of a single Orbitrap MS scan (350–1800 m/z, maximal injection time of 50 ms, an Automated Gain Control (AGC) set to a maximum of 1×10$^6$ ions and a resolution of 120k) every 3 s and Orbitrap MS/MS HCD scans of precursors (NCE 28% with 5% Stepping, maximal injection time of 60 ms, an AGC set to a maximum of 1×10$^5$ ions and a resolution of 15k). Scans containing the oxonium ions (204.0867; 138.0545 or 366.1396 m/z) triggered three additional product-dependent MS/MS scans[73] of potential glycopeptides; an Orbitrap EThcD scan (NCE 15%, maximal injection time of 250 ms, AGC set to a maximum of 2×10$^5$ ions with a resolution of 30k and using the extended mass range setting to improve the detection of high mass glycopeptide fragment ions[74]); an ion trap CID scan (NCE 35%, maximal injection time of 40 ms, an AGC set to a maximum of 5×10$^4$ ions) and a stepped collision energy HCD scan (using NCE 35% with 8% Stepping, maximal injection time of 150 ms, an AGC set to a maximum of 2×10$^5$ ions and a resolution of 30k).

**Glycopeptide identifications using Byonic**. Raw data files were processed using Byonic v3.5.3 (Protein Metrics Inc.[75]). Tryptic samples were searched with a n-ragged semi-tryptic specificity allowing a maximum of two missed cleavage events while Pepsin and Thermolysin samples were searched with non-specific specificity. Carbamidomethyl was set as a fixed modification of cysteine while oxidation of methionine was included as a variable modification. The Burkholderia glycans HexHexNAc$_2$ (elemental composition: $C_{22}O_{15}H_{36}N_2$, mass: 568.2115) and Suc-HexHexNAc$_2$ (elemental composition: $C_{26}O_{18}H_{40}N_2$, mass: 668.2276) were included as variable modifications at Serine and Threonine residues. K56-2 samples were searched against the K56-2 proteome[76] (Uniprot accession: UP000011196, 7467 proteins) while H111 samples were searched against the H111 proteome[41] (Uniprot accession: UP000245426, 8111 proteins). A maximum mass precursor tolerance of 5 ppm was allowed while a mass tolerance of up to 10 ppm was set for HCD fragments and 20 ppm for EThcD fragments. Separate datasets from the same strain were combined using R and only glycopeptides with a Byonic score >300 used for further analysis. This score cut-off is in line with previous reports highlighting that score thresholds greater than at least 150 are required for robust tryptic glycopeptide assignments within Byonic[77,78]. Manual inspection of glycopeptides to assess the correctness of assignments was undertaken using the guidelines of Chen et al.[79] with the additional requirement that HCD glycopeptide spectra should contain evidence for glycan fragments such as oxonium ions or the presence of Y$_0$ to Y$_2$ ions. Glycosylation sites were defined as localised if EThcD scans enabled the unambiguous assignment to a specific Serine/Threonine residue based on c and z ions or a HCD scan contained only a single Serine/Threonine residue. Partial localisation is defined as EThcD spectra containing fragmentation information which allows the ruling out of potential Serine/Threonine residues yet does not provide evidence enabling the exact site of glycosylation to be assigned. All glycopeptide spectra satisfying the correctness of assignments guidelines outlined above, as well as spectra which provided either partial or complete localisation of glycosylation sites, are provided within Supplementary Data 3.

**Analysis of glycosylation sites using O-Pair**. Automated glycosylation site analysis was undertaken using O-Pair within MetaMorpheus (non-public release version MM0523[54]). Glycopeptide enriched samples were first processed using the

Freestyle Viewer (1.7 SP1, Thermo Fisher Scientific) to remove ion-trap CID scans and then searched allowing for a maximum of 4 glycans. The *Burkholderia* glycans were defined as HexHexNAc$_2$ (elemental composition: $C_{22}O_{15}H_{36}N_2$, mass: 568.2115) or Suc-HexHexNAc$_2$ (elemental composition: $C_{26}O_{18}H_{40}N_2$, mass: 668.2276). K56-2 samples were searched against the K56-2 proteome[76] (Uniprot accession: UP000011196) while H111 samples were searched against the H111 proteome[41] (Uniprot accession: UP000245426). Tryptic samples were searched using the default settings while Pepsin and Thermolysin samples were searched with non-specific specificity allowing 5 to 55 amino acids and the search partitioned into 100 partitions. Analysis of the tryptic glycoproteome samples of *B. pseudomallei* K96243; *B. multivorans* MSMB2008; *B. dolosa* AU0158; *B. humptydooensis* MSMB43; *B. ubonensis* MSMB22; *B. anthina* MSMB649; *B. diffusa* MSMB375 and *B. pseudomultivorans* MSMB2199 (PRIDE ProteomeXchange Consortium accession: PXD018587) were undertaken as above using the proteome databases as outlined in[13]. Only class "level 1" glycopeptides with a Q-value less than 0.01 and a site-specific probability of >0.75 were considered as localised and used for further analysis.

**Digestion of whole-cell samples for proteomic comparisons**. Dried protein pellets of H111 strains were resuspended in 6 M urea, 2 M thiourea, 40 mM $NH_4HCO_3$ and reduced/alkylated prior to digestion with Lys-C then trypsin overnight as described above. Digested samples were acidified to a final concentration of 0.5% formic acid and desalted with home-made high-capacity StageTips composed of 1 mg Empore™ C18 material (3 M) and 5 mg of OLIGO R3 reverse phase resin (Thermo Fisher Scientific) as described[72,80]. Columns were wet with Buffer B (0.1% formic acid, 80% acetonitrile) and conditioned with Buffer A* prior to use. Acidified samples were loaded onto conditioned columns, washed with 10 bed volumes of Buffer A*, and bound peptides were eluted with Buffer B, before being dried via vacuum centrifugation at room temperature and stored at −20 °C.

**LC-MS analysis of H111 proteome samples**. Stagetip cleaned-up H111 proteome samples were resuspended in Buffer A* and separated using a two-column chromatography set up comprised of a PepMap100 C18 20 mm×75 μm trap and a PepMap C18 500 mm×75μm analytical column (Thermo Scientific). Samples were concentrated onto the trap column at 5 μl/min with Buffer A for 5 min and infused into an Orbitrap Elite™ Mass Spectrometer (Thermo Scientific) at 300 nl/min via the analytical column using an Dionex Ultimate 3000 UPLC (Thermo Scientific). The peptides were separated using a 125-min gradient altering the buffer composition from 1% buffer B to 23% B over 95 min, then from 23% B to 40% B over 10 min, then from 40% B to 100% B over 5 min, the composition was held at 100% B for 5 min, and then dropped to 3% B over 2 min and held at 3% B for another 8 min. The Orbitrap Elite™ was operated in a data-dependent mode automatically switching between the acquisition of a single Orbitrap MS scan (300–1650 m/z, maximal injection time of 50 ms, an AGC set to a maximum of $1\times10^6$ ions and a resolution of 60k) followed by 20 data-dependent ion-trap CID MS-MS events (NCE 30%, maximal injection time of 80 ms, an AGC set to a maximum of $2\times10^4$ ions).

**In-gel digestion of DsbA1$_{Nm}$-his$_6$ within Burkholderia lysates**. Whole-cell lysates of induced *Burkholderia* strains prepared as above for immunoblotting were separated on pre-cast 4–12% gels then fixed in fixative buffer (10% methanol, 7% acetic acid) for 1 h before being stained with Coomassie G-250 overnight. The region corresponding to ~25–35 kDa was then excised and processed as previously described (38). Briefly, the excised gel regions were sectioned into ~2 mm$^2$ pieces then destained in a solution of 100 mM $NH_4HCO_3$/50% ethanol for 15 min at room temperature with shaking. Destaining was repeated twice to ensure the removal of excess Coomassie. Destained bands were dehydrated with 100% ethanol for 5 min and then rehydrated in 50 mM $NH_4HCO_3$ containing 10 mM DTT. Samples were reduced for 60 min at 56 °C with shaking and washed twice in 100% ethanol for 10 min to remove DTT. Reduced ethanol washed samples were sequentially alkylated with 55 mM of iodoacetamide in 50 mM $NH_4HCO_3$ in the dark for 45 min at room temperature. Alkylated samples were then washed with Milli-Q water followed by 100% ethanol twice for 5 min to remove residual iodoacetamide then vacuum-dried for 10 min. Alkylated samples were then rehydrated with 20 ng/μl trypsin (Promega) in 50 mM $NH_4HCO_3$ at 4 °C for 1 h. Excess trypsin was removed, the gel pieces covered in 40 mM $NH_4HCO_3$, and incubated overnight at 37 °C. Peptides were concentrated and desalted using C18 stage tips (64, 65) before analysis by LC-MS.

**LC-MS analysis of DsbA1$_{Nm}$-his$_6$ glycopeptides**. Stagetip cleaned-up samples were resuspended in Buffer A* and separated using a two-column set up as described above, coupled to a Orbitrap Lumos™ Mass Spectrometer equipped with a FAIMS Pro interface (Thermo Fisher Scientific). 125-min gradients were run for each sample altering the buffer composition from 3% Buffer B to 28% B over 95 min, then from 28% B to 40% B over 10 min, then from 40% B to 80% B over 7 min, the composition was held at 80% B for 3 min, and then dropped to 3% B over 1 min and held at 3% B for another 9 min. The Lumos™ Mass Spectrometer was operated in a stepped FAIMS data-dependent mode at three different FAIMS

CVs −25, −45 and −65 as previously described[49]. For each FAIMS CV a single Orbitrap MS scan (350–2000 m/z, maximal injection time of 50 ms, an AGC of maximum of $1\times10^6$ ions and a resolution of 120k) was acquired every 2 s followed by Orbitrap MS/MS HCD scans of precursors (NCE 30%, maximal injection time of 80 ms, an AGC set to a maximum of $1\times10^5$ ions and a resolution of 15k). HCD scans containing the oxonium ions (204.0867; 138.0545 or 366.1396 m/z) triggered an additional Orbitrap EThcD scan, an ion-trap CID scan and a Orbitrap HCD scan of potential glycopeptides with scan parameters described above. For parallel reaction monitoring (PRM) experiments, 65-min gradients were run for each sample altering the buffer composition from 3% buffer B to 28% B over 35 min, then from 28% B to 40% B over 10 min, then from 40% B to 80% B over 7 min, the composition was held at 80% B for 3 min, and then dropped to 3% B over 1 min and held at 3% B for another 9 min. The Lumos™ Mass Spectrometer was operated at a FAIMS CV of −25 in a data-dependent mode automatically switching between the acquisition of a single Orbitrap MS scan (350–2000 m/z, maximal injection time of 50 ms, an AGC set to a maximum of $1\times10^6$ ions and a resolution of 60k) every 3 s followed by ion-trap EThcD MS2 events (NCE 25%, maximal injection time of 200 ms, an AGC set to a maximum of $6\times10^4$ ions) of precursors and then a Orbitrap EThcD PRM scan (NCE 25%, maximal injection time of 450 ms, an AGC set to a maximum of $2\times10^5$ ions) of the m/z 1547.77, 1581.12 and 1543.10 which corresponds to the +3 charge states of the HexHexNAc$_2$ glycosylated [23]VQTSVP ADSAPAATAAAAPAGLVEGQNYTVLANPIPQQQAGK[64]; the Suc-HexHexNAc$_2$ glycosylated [23]VQTSVPADSAPAAT̲AAAAPAGLVEGQNYTVLANPIPQQ QAGK[64] and the HexHexNAc$_2$ glycosylated [23]VQTSVPADSAPAASAAAAPAGL VEGQNYTVLANPIPQQQAGK[64] peptides with the site of glycosylation within these peptides underlined.

**Proteomic analysis**. H111 Proteome and in-gel datasets were processed using MaxQuant (v1.5.5.1 or 1.6.3.4.[81]). The H111 proteome dataset was searched against the H111 proteome[41] (Uniprot accession: UP000245426) and *B. cenocepacia* strain J2315 (Uniprot accession: UP000001035, 6993 proteins) to enable the matching of J2315 gene accessions. In-gel digests were searched against either *B. cenocepacia* strain J2315 (Uniprot accession: UP000001035), *B. ubonensis* MSMB22 (Burkholderia Genome Database[55], Strain number: 3404) or *B. humptydooensis* MSMB43 (Burkholderia Genome Database[55], Strain number: 4072) depending on the sample type and a custom database of DsbA1$_{Nm}$-his$_6$ containing the desired point mutations at position 31 and 36 within Uniprot entry Q9K189. All searches were undertaken using "Trypsin" enzyme specificity with carbamidomethylation of cysteine as a fixed modification. Oxidation of methionine and acetylation of protein N-termini were included as variable modifications and a maximum of 2 missed cleavages allowed. For in-gel samples HexHexNAc$_2$ (elemental composition: $C_{22}O_{15}H_{36}N_2$, mass: 568.2115) and Suc-HexHexNAc$_2$ (elemental composition: $C_{26}O_{18}H_{40}N_2$, mass: 668.2276) were also included as variable modifications. To enhance the identification of peptides between samples, the Match between Runs option was enabled with a precursor match window set to 2 min and an alignment window of 10 min. For label free quantitation (LFQ) the MaxLFQ option in Maxquant[82] was enabled. The resulting outputs were processed within the Perseus (v1.5.0.9)[83] analysis environment to remove reverse matches and common protein contaminants prior to further analysis. For LFQ comparisons missing values were imputed based on the observed total peptide intensities with a range of 0.3σ and a downshift of 2.5σ using Perseus. Enrichment analysis was undertaken using Fisher exact tests in Perseus with PglL altered proteins defined as those proteins which were previously reported by Oppy et al. as differentially altered in K56-2 Δ*pglL* compared to K56-2 WT[5]. Fisher exact tests in Perseus was undertaken allow a 5% FDR. To compare the relative abundance of glycosylated and non-glycosylated peptides from DsbA1$_{Nm}$-his$_6$ point mutants the area under the curve of peptides were extracted using the FreeStyle viewer and the resulting data provided within the Supplementary document. Statistical analysis of the area under the curve was undertaken in Prism (version 7.0e) using a two-sided t-test.

**Visualisation of glycoproteome and proteome datasets**. Data visualisation was undertaken using ggplot2[84] within R with all scripts included in the PRIDE uploaded datasets. To aid in the analysis of the MS/MS data the Interactive Peptide Spectral Annotator[85] (http://www.interactivepeptidespectralannotator.com/PeptideAnnotator.html) was used.

**Bioinformatic analysis of glycoproteins and glycosylation sites**. Draft and complete genome sequences of *B. cenocepacia* ($n = 294$) and complete genome sequences of other *Burkholderia* species, subsetted to contain a maximum of 20 of any individual species ($n = 174$), were obtained from the *Burkholderia* database[55]. Isolates were screened for the presence of the glycosylated motif containing genes from *B. cenocepacia* J2315 using an 80% identity and length BlastN thresholds using screen_assembly3.py v1.2.7[86]. Gene hits were translated to protein sequences and the translated hits screened for the presence of the motif using seqkit locate[87], allowing for up to 4 mismatched amino acids (> 80% conservation of the 21 amino acid motif). The identity and coverage of the genes at 80% identity as well as the motif coverage at 80% identity were visualised using ggplot2. Sequence logos were generated using ggseqlogo[88]. To compare glycoprotein homologues and pglL

sequences between *Burkholderia* species the *Burkholderia* Orthologous Groups information provided by the *Burkholderia* database was used[55].

**Statistics and Reproducibility**. Statistical analyses of biological samples were undertaken on a minimum of three biological replicates for Western blotting or Proteomic analysis. A biological replicate is defined as separately grown cultures treated with a given induction or growth condition. All raw uncropped and protein marker associated Western blotting images are provided within Supplementary Fig. 19.

**Reporting summary**. Further information on research design is available in the Nature Research Reporting Summary linked to this article.

## Data availability

Mass spectrometry data (Raw data files, Byonic/Maxquant/O-pair search outputs, R Scripts and output tables) have been deposited into the PRIDE ProteomeXchange Consortium repository[89,90] (https://www.ebi.ac.uk/pride/archive/). The glycoproteomic datasets are available with the identifier: PXD024090; the H111 proteome analysis is available with the identifier: PXD023755; the DsbA1$_{Nm}$-his$_6$ (*B. cenocepacia* K56-2) associated analysis is available with the identifier: PXD023955; the DsbA1$_{Nm}$-his$_6$ (*B. humptydooensis* MSMB43 and *B. ubonensis* MSMB22) associated analysis is available with the identifier: PXD024056.

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

## Acknowledgements

N.E.S is supported by an Australian Research Council Future Fellowship (FT200100270) and an ARC Discovery Project Grant (DP210100362). M.R.D is supported by a University of Melbourne CR Roper fellowship. We thank the Melbourne Mass Spectrometry and Proteomics Facility of The Bio21 Molecular Science and Biotechnology Institute for access to MS instrumentation and Byonic. We thank Lei Lu, Nick Riley and Lloyd M. Smith for providing the custom version of O-Pair for the analysis of *Burkholderia* glycopeptides. pSCrhaB2 was a gift from Miguel Valvano (Addgene plasmid # 113634; http://n2t.net/addgene:113634; RRID:Addgene_113634)

## Author contributions

A.J.H. undertook the bioinformatic analysis of glycoproteins and glycosylation sites conserved across Burkholderia species. J.M.L aided in the preparation of the manuscript and its figures. M.R.D oversaw bioinformatic analysis and aided in the preparation of the manuscript. N.E.S undertook all proteomic and molecular biology studies as well as wrote the initial manuscript draft.

## Competing interests

The authors declare no competing interests.
