## [Transparent Peer Review File · Communications Biology]

Reviewers' comments:

Reviewer #1 (Remarks to the Author):

This work by Hayes et al is a technically impressive, complete, and convincing body of work that provides some surprising novel insights into the O-glycoproteome of the diverse and important Burkholderia genus. In particular, the extreme preference for PglL oligosaccharyltransferases to glycosylate serine over threonine is consistent with the literature, but to my knowledge has not previously been documented as systematically as in this study. The strong conservation of glycosylation sites and glycoproteins across Burkholderia is also noteworthy, in its implications for understanding the fundamental glycobiology of the genus, and because of the importance of O-glycosylation for the virulence of these bacteria.

The experimental, technical, and statistical aspects of the work are clearly described and all appear to be appropriate.

I have several suggestions that I think would improve the clarity of the manuscript that the authors may choose to address.

- 1) "oligosaccharidetransferase". Oligosaccharyltransferase is more standard.
- 2) "D/E-X-N-X-S/T". It should be noted that X1 is not necessarily the same as X2, and neither can be P.
- 3) It would be helpful if a brief description of H111 and K56-2 could be provided in the introduction, overviewing their genomic similarity and outlining any known or expected differences in their biology, specifically related to glycosylation.
- 4) The authors note that coverage of the glycoproteome was improved by using separate digests with the complementary proteases trypsin, thermolysis, and pepsin. It would be interesting if more detail could be included describing and comparing the performance of these enzymes for glycopeptide identification.
- 5) Over 65% of glycoproteins were identified in both K56-2 and H111. Are the glycosylation sites identified in high quality glycopeptides but unique to one strain also present in the other strain, even if they are not identified as glycopeptides? That is, can the differences between the identified glycopeptides in each strain potentially be explained by differences in protein sequence, glycosylation occupancy, or analytical detection?
- 6) Figure 1C. It would be helpful to note that anti-RNA pol is used as a loading control. RNA pol appears to show a difference in MW depending on the presence of PglL. Can this be explained?
- 7)
"23VQTSVPADSAPAASAAAAPAGLVEGQNYTVLANPIPQQQAGK64"
"23VQTSVPADSAPAATAAAAAPAGLVEGQNYTVLANPIPQQQAGK64"
It would be helpful to label or annotate (e.g. with numbering or in bold) the potentially glycosylated S26, S31, and S36; and the site-directed mutated variant T36.
- 8) In this study glycopeptides were identified after enrichment. In the discussion it is mentioned that glycosylation can be regulated or affected by factors such as growth conditions. The immunoblots in Figure 1C also suggest that BCAL2466 is partially glycosylated, while BCAL2345 is completely glycosylated. Can you comment on the quantitative occupancy of the O-glycosylation events described in this study, and if high-occupancy sites have specific sequence characteristics?

I congratulate the authors on an excellent study.

Benjamin L. Schulz

Reviewer #2 (Remarks to the Author):

This is a generally well written ms identifying an interesting aspect of bacterial O-linked protein glycosylation in Burkholderia species – namely a strong bias for serine over threonine at the glycosylation site. This is certainly a novel finding and will be of interest to the glycobiology field particularly those studying bacterial protein glycosylation. Indeed, this preference for serine may be a more general feature of PgL-type oligosaccharyltransferases in other important bacterial pathogens. This finding will also inform work to develop these protein glycosylation systems as bioengineering tools for production of glycoconjugate vaccines. There were further data presented indicating conservation of the targets of O-linked protein glycosylation among Burkholderia species. Overall, the data are certainly convincing, combining global glycoproteomics with more targeted analyses of specific proteins, with both supporting the strong preference of the oligosaccharyltransferase for serine over threonine. The experiments are well described with a detailed methods section.

General points:

Line 442 – It would be useful to explain how you got the figure of >65% for glycoproteins present in both strains from Figure 1B; I got that 63% of glycoproteins in H1111 were also found in K56-2. You might also consider clarifying the wording here to make sure the reader is made aware of exactly where these figures derive from.

Figure 1C – Could you explain the differing mobilities of RNAPol (presumably a loading control though this is not made explicit in legend) in the 2466 blot? For these data it looks like RNAPol protein is modified in a PgL dependent manner! This needs to be addressed.

Figure 3A – There are faint bands above all the single intense bands in this figure and so I would be very careful about inferring that this upper band in the final lane is due to glycosylation. I don't think you can ignore the other bands. Could these low intensity upper bands represent the protein with signal peptide still attached?

In Figures 2B, 3A and 5BC the DsbA1A36 protein is slightly more mobile than DsbA1T36. Why is this? It certainly seems consistent but is not commented on and should be or even investigated further.

In Figure 2A why is the threonine present in the logo when the experimental data do not support this?

Specific points:

Line 46 "installation of glycans" – unusual terminology?

52 "commonality" needs explaining to clarify

62 "unique" consider word choice here and intended meaning

86 "predominately" predominantly?

107 "proteins" to protein

190 missing full stop

229/231 "a" to "an"

259 "deemed correct" consider rewording to clarify

319 "Iodacetamide" has initial capital – other chemicals don't

376 "contaminates" to contaminants?

384 "statistically" to statistical?

421 two full stops

427/437 insert "respectively" as appropriate

441 "predominately" again

446 How were these two proteins chosen?

450 Expression of the 2466 and 2345 proteins in the delta pglL strain results in increased gel mobility rather than "reduced gel mobility" as stated. It would also be useful to mention that for 2466 both unglycosylated and glycosylated forms are observed in the PglL competent strain. Some readers may need this help with interpretation here.

Reviewer #3

1. The last paragraph (line 628) of the discussion needs a little clarification, I am not sure I understand the final conclusions as written.

**Response to referees**

Reviewers' comments:

Reviewer #1 (Remarks to the Author):

This work by Hayes et al is a technically impressive, complete, and convincing body of work that
provides some surprising novel insights into the O-glycoproteome of the diverse and important
Burkholderia genus. In particular, the extreme preference for PglL oligosaccharyltransferases to
glycosylate serine over threonine is consistent with the literature, but to my knowledge has not
previously been documented as systematically as in this study. The strong conservation of
glycosylation sites and glycoproteins across Burkholderia is also noteworthy, in its implications
for understanding the fundamental glycobiology of the genus, and because of the importance of
O-glycosylation for the virulence of these bacteria.

The experimental, technical, and statistical aspects of the work are clearly described and all
appear to be appropriate.

I have several suggestions that I think would improve the clarity of the manuscript that the
authors may choose to address.

1) "oligosaccharidetransferase". Oligosaccharyltransferase is more standard.

We thank the reviewer for pointing out this oversight. We have now changed
oligosaccharidetransferase to oligosaccharyltransferase throughout the manuscript.

2) "D/E-X-N-X-S/T". It should be noted that X1 is not necessarily the same as X2, and neither can
be P.

We have corrected the definition of the glycosylation sequon and updated the description of the
Campylobacter glycosylation sequon to:

Line 57 to 59 "i) glycosylation events within the *Campylobacter* genus occurring within the
glycosylation sequon, D/E-X₁-N-X₂-S/T (where neither X₁ or X₂ can be proline)¹⁹, which restricts
the possible glycosylation sites within a protein"

3) It would be helpful if a brief description of H111 and K56-2 could be provided in the
introduction, overviewing their genomic similarity and outlining any known or expected
differences in their biology, specifically related to glycosylation.

We thank the reviewer for this suggestion and agree that an introduction to these two strains
would be helpful to readers. We have now added the following paragraph to the introduction.

Line 92 to 103 "The opportunistic pathogen *B. cenocepacia* is associated with life-threatening
infections in people with cystic fibrosis (CF)³⁵. While ubiquitous in the environment not all *B.*
*cenocepacia* strains are associated with human disease with the phylogenetic grouping known as

the IIIA genomovar overrepresented in CF infections ³⁶⁻³⁸. Within the IIIA genomovar, the K56-2
 ³⁹ and H111 ⁴⁰ strains have been established as the most widely used models of *B. cenocepacia*
 pathogenesis. While both strains were isolated from CF patients, only K56-2 is a member of the
 ET12 lineage, a highly transmissible lineage associated with high mortality rates ^{42,43}. The absence
 of key genetic elements in H111, such as the low-oxygen-activated locus (*lxa*) ⁴⁴ and the ccilR
 quorum-sensing system ^{45,46}, has been suggested to account for the notable differences in
 virulence traits ⁴⁷ and quorum-sensing ^{48,49} observed between K56-2 and H111. This genetic
 diversity between IIIA genomovar isolates makes understanding multiple strains essential for
 identifying core features shared across the majority of pathogenic *B. cenocepacia* isolates.”

 4) The authors note that coverage of the glycoproteome was improved by using separate digests
 with the complementary proteases trypsin, thermolysis, and pepsin. It would be interesting if
 more detail could be included describing and comparing the performance of these enzymes for
 glycopeptide identification.

 We thank the reviewer for this suggestion, we have now undertaken further analysis comparing
 the performance of these proteases for glycopeptide identification. Comparing the scores and
 localisation information reveals that both thermolysis and pepsin typically have lower Byonic
 scores than trypsin yet the percentage of unique glycopeptides which enabled the localisation of
 sites by each protease were similar. This is consistent with our observation that score alone does
 not predict glycosylation localisation and further highlights the benefits of using multiple
 enzymes to improve the coverage of glycoproteomes (Reviewer Comment Figure 1).

Review Comment Figure 1: Glycopeptide identification performance using different proteases.
 Curated glycopeptide datasets (Supplementary table 1 and 2) demonstrate differences in the

score distributions of each protease yet reveal each protease resulted in a similar rate of
glycosylation site localisation.

This analysis has now been included within the manuscript as Supplementary Figure 3 and the
following statement added to the results section:

Line 452 to 456 “Comparing the observed score distributions for each proteases revealed that
for both H111 and K56-2 Thermolysin and Pepsin glycopeptides typically possessed lower and
narrower score distributions, yet still enabled the localisation of glycosylation sites at a similar
frequency as Trypsin (>50% of unique glycopeptides localised, Supplementary Figure 3A to D).”

5) Over 65% of glycoproteins were identified in both K56-2 and H111. Are the glycosylation sites
identified in high quality glycopeptides but unique to one strain also present in the other strain,
even if they are not identified as glycopeptides? That is, can the differences between the
identified glycopeptides in each strain potentially be explained by differences in protein
sequence, glycosylation occupancy, or analytical detection?

The reviewer raises an excellent point about the cause of the partial overlap between the
glycoproteome of K56-2 and H111. We agree that multiple factors may be contributing to this
including differences in protein sequences, glycosylation occupancy, protein abundance, or the
detection/filtering of glycopeptides and address each of these potential sources below.

Differences in protein sequences: As highlighted within Figure 4 and Supplementary Figures 9 to
14 our analysis of 70 glycoproteins with localised glycosylation sites revealed that the majority of
glycosylation sites were highly conserved across examined *B. cenocepacia* genomes. It is
noteworthy that both H111 and K56-2 were included in this analysis, demonstrating the majority
of glycosylation sites are both conserved and are identical between the two strains. This suggests
the absence of Serine residues between these strains does not contribute to this partial overlap.

Differences in the detection or filtering of glycopeptides: One potential source of this partial
overlap could be the inadvertent removal of correct, albeit low scoring, glycopeptides during the
filtering of glycopeptides identified by Byonic. To examine this possibility, we assessed the
overlap in the peptide sequences within our curated glycopeptide data to all possible
glycopeptides identified at a 1% FDR between strains (Review Comment Figure 2). Even allowing
lower scoring peptides (Byonic scores >140) we see the overlap is modest at best (between 37-
49%). It is important to note that at the peptide level this overlap does not perfectly reflect
protein level overlap due to different peptides sequences derived from the same protein being
observed between strains.

Review Comment Figure 2: Overlap within curated glycopeptide datasets and with all Byonic identified glycopeptide within the strains.

Differences in protein abundance: Leveraging our proteomic datasets we have now assessed if the difference in the glycoproteins observed within H111 and K56-2 could be driven by differences in protein expression. Across these proteomes the majority of glycoproteins (66 glycoproteins) are observed in both strains with an additional 14 proteins observed within a single strain (Review Comment Figure 3). This data supports that at least 66 out of the 98 glycoproteins are expressed in both strains supporting that the partial overlap cannot be explained simply by the lack of expression of these glycoproteins.

Review Comment Figure 3: Overlap of glycoproteins within the observed H111 and K56-2 proteomes. Now provided as Supplementary Figure 18.

Based on this analysis the differences in the glycoproteomes observed between strains is unlikely to be due to filtering of glycopeptides during data handling, genetic variation between strains, or the lack of expression of these glycoproteins within strains. This suggests that differences in glycopeptide sampling or occupancy between strains may be driving this partial overlap yet it's important to note we lack evidence for either scenario. Despite this we agree that it's important to highlight these two scenarios to readers and we have added the following statement to the discussion:

Line 642 to 649 “Although highly conserved, it is important to note at the glycoproteome level
 we observed only 47 of the 98 glycoproteins identified within both strains (Figure 1B).
 Examination of the proteomes of H111 and K56-2 supports that this modest overlap is not due
 to the lack of expression of these glycoproteins, as at least 66 of these glycoproteins are
 expressed in both strains (Supplementary Figure 18). Rather, this overlap supports that the
 differences in the observable glycoproteome are driven by either under-sampling of the
 glycoproteome due to the low abundance of these glycoproteins, or differences in glycosylation
 occupancy between strains.”

 6) Figure 1C. It would be helpful to note that anti-RNA pol is used as a loading control. RNA pol
 appears to show a difference in MW depending on the presence of PglL. Can this be explained?

We thank the reviewer for highlighting these points. We have now added the statement “with
 anti-RNA pol westerns included as loading controls” to the figure legend of Figure 1 (Line 812 and
 813) to clearly highlight that these are loading controls. In our hands, using pre-cast gels, we have
 noted that sometime small wobbles in mobility can be observed between wells. The initial RNA
 pol western provided within Figure 1C was used as it was generated by re-probing the initial
 BCAL2466-his blot and although it shows consistent loading the wobble is not ideal. To address
 this issue, we have repeated the westerns for both BCAL2466-his and BCAL2345-his and included
 additional protein loading controls in the form of Coomassie stained gels (Review Comment
 Figure 4). As can be seen comparing the RNA pol western between BCAL2466-his and BCAL2345-
 his reveals no change in mobility between the glycosylation null strain K56-2 $\Delta pglL$ compared to
 the K56-2 WT. Figure 1C has now been updated with these more ideal westerns.

 Review Comment Figure 4: Western blotting of two potential glycoproteins, His₆ tagged
 BCAL2466 and BCAL2345, reveals alterations in protein banding when expressed within the

glycosylation null strain K56-2 Δ pglL compared to the K56-2 WT with anti-RNA pol westerns and
Coomassie G-250 gels included as loading controls.

7)"23VQTSVPADSAPAASAAAAPAGLVEGQNYTVLANPIPQQQAGK64"

"23VQTSVPADSAPAATAAAAAPAGLVEGQNYTVLANPIPQQQAGK64"

It would be helpful to label or annotate (e.g. with numbering or in bold) the potentially
glycosylated S26, S31, and S36; and the site-directed mutated variant T36.

We thank the reviewer for their suggestion, we have now underlined the glycosylation sites
within peptides to improve the clarity of which amino acid is being referred to.

8) In this study glycopeptides were identified after enrichment. In the discussion it is mentioned
that glycosylation can be regulated or affected by factors such as growth conditions. The
immunoblots in Figure 1C also suggest that BCAL2466 is partially glycosylated, while BCAL2345 is
completely glycosylated. Can you comment on the quantitative occupancy of the O-glycosylation
events described in this study, and if high-occupancy sites have specific sequence characteristics?

The reviewer raises an excellent question which is an active area of research within the lab. The
reviewer is correct that differences in glycosylation occupancy can clearly be seen between
BCAL2466 and BCAL2345, yet we currently lack a complete understanding of what is driving this.
For the majority of glycoproteome data generated here we have no information about the
relative occupation rates beyond if the sequence is glycosylated. Moving forward a key goal of
the lab is to track occupancy by measuring both the glycosylated and non-glycosylated version of
peptides and total protein amounts using TMT labelling to dissect the specific sequence
characteristics which control glycosylation.

To make it clear to readers that quantitative occupancy of O-glycosylation events is unknown for
the majority of glycosylation sites we have added the following statement to the discussion:

Line 626 to 631 "Although Serines are the preferred glycosylated residue across *Burkholderia*
glycoproteins, it is unclear if all glycosylated Serine's are modified to a high level of occupancy.
Our western blot analysis of His₆ tagged BCAL2466 and BCAL2345 (Figure 1C) supports that
differences in glycosylation occupancy do exist, yet whether specific sequence or protein
characteristics predict this efficiency are unclear. Together, this suggests further studies are
required to understand the properties which promote occupancy at specific sites."

I congratulate the authors on an excellent study.

Benjamin L. Schulz

Reviewer #2 (Remarks to the Author):

This is a generally well written ms identifying an interesting aspect of bacterial O-linked protein
glycosylation in Burkholderia species – namely a strong bias for serine over threonine at the
glycosylation site. This is certainly a novel finding and will be of interest to the glycobiology field
particularly those studying bacterial protein glycosylation. Indeed, this preference for serine may
be a more general feature of PglL-type oligosaccharyltransferases in other important bacterial
pathogens. This finding will also inform work to develop these protein glycosylation systems as
bioengineering tools for production of glycoconjugate vaccines. There were further data
presented indicating conservation of the targets of O-linked protein glycosylation among
Burkholderia species. Overall, the data are certainly convincing, combining global
glycoproteomics with more targeted analyses of specific proteins, with both supporting the
strong preference of the oligosaccharyltransferase for serine over threonine. The experiments
are well described with a detailed methods section.

General points:

Line 442 – It would be useful to explain how you got the figure of >65% for glycoproteins present
in both strains from Figure 1B; I got that 63% of glycoproteins in H1111 were also found in K56-
2. You might also consider clarifying the wording here to make sure the reader is made aware of
exactly where these figures derive from.

The figure of 65% was calculated by averaging the number of glycoproteins observed in both
strains $(47/70 + 47/75)/2 = \sim 65\%$

For simplicity we have changed >65% to $\sim 65\%$ within the manuscript (Line 457)

Figure 1C – Could you explain the differing mobilities of RNAPol (presumably a loading control
though this is not made explicit in legend) in the 2466 blot? For these data it looks like RNAPol
protein is modified in a PglL dependent manner! This needs to be addressed.

We thank the reviewer for highlighting this irregularity, we have addressed this comment above
and have demonstrated that this difference in mobility is an artefact.

Figure 3A – There are faint bands above all the single intense bands in this figure and so I would
be very careful about inferring that this upper band in the final lane is due to glycosylation. I don't
think you can ignore the other bands. Could these low intensity upper bands represent the
protein with signal peptide still attached?

The reviewer raises an excellent point that the upper band that is more abundant within the
overexpression complement strain may not be solely due to glycosylation but an alternative
modification such as inefficient cleavage of the signal tag. We are fortunate that for DsbA1_{Nm}-
his₆ the N-terminal is extremely amenable to analysis with Trypsin (with the terminal sequence

¹MKSRHLALGV ¹¹AALFALAACD ²¹SKVQTSVPAD³⁰) enabling us to monitor the N-termini within our
proteomic dataset. From our analysis of DsbA1_{Nm}-his₆ forms (Supplementary table 7) we note we
are unable to detect any peptides preceding V²³. This supports that the signal tag on the observed
DsbA1_{Nm}-his₆ has been removed but agree that our language should be more cautious especially
due to the low abundance of the glycosylated form of T³⁶ DsbA1_{Nm}-his₆. In light of this we have
modified our description of this from:

“Yet, in contrast to K56-2 WT, a faint band potentially corresponding to glycosylated DsbA1_{Nm}-
his₆ T³⁶ was observed (Figure 3A). Targeted MS analysis confirms the glycosylation of T³⁶ in
DsbA1_{Nm}-his₆ T³⁶ when expressed in K56-2 Δ *pglL* AmrAB::*S7-pglL-his* (Figure 3B and C).”

to

Line 531 to 534 “Yet, in contrast to K56-2 WT an additional faint band is observable within
DsbA1_{Nm}-his₆ T³⁶ (Figure 3A), consistent with the presence of low abundance glycosylated
DsbA1_{Nm}-his₆ T³⁶. Targeted MS analysis confirms the glycosylation of T³⁶ in DsbA1_{Nm}-his₆ T³⁶ when
expressed in K56-2 Δ *pglL* *amrAB*::*S7-pglL-his*₁₀ (Figure 3B and C).”

In Figures 2B, 3A and 5BC the DsbA1A36 protein is slightly more mobile than DsbA1T36. Why is
this? It certainly seems consistent but is not commented on and should be or even investigated
further.

We thank the reviewer for highlighting this mobility difference, we are unsure what is causing
this small but reproducible shift in DsbA1_{Nm}-his₆ A³⁶ with our proteomic and sequencing results
ruling out an unexpected modification of DsbA1_{Nm}-his₆ outside of the expected A³⁶. To alert the
readers to this consistent change we have added the following sentence to the results section:

Line 517 to 519 “Curiously DsbA1_{Nm}-his₆ A³⁶ leads to a slight but reproducible increased mobility
compared to other point mutants yet our proteomic and sequencing result support the
correctness of this construct, and as such the cause of this shift remains unknown.”

Proteomics

In Figure 2A why is the threonine present in the logo when the experimental data do not support
this?

We initially included the assigned Threonine site within Figure 2A as this site initially met our
inclusion criteria and was instrumental in leading to the re-analysis of BCAM0996 T¹⁵⁹. We
appreciate the reviewer’s comment that strictly speaking this site is known to be incorrect and
therefore should be removed. We have now updated Figure 2A to remove the incorrectly
assigned Threonine site (shown below) and moved the original version of the sequon analysis to
the Supplementary Figure 4.

**Updated Figure 2. O-linked glycosylation predominantly occurs on Serine residues across the**
***B. cenocepacia* glycoproteome.**

The results section has been rewritten to reflect this alteration to the paper as follows:

Line 488 to 500 “Surprisingly this analysis suggested a single Threonine residue, T¹⁵⁹ within
BCAM0996, was modified within H111. Due to the discordance of this assignment with respects
to all other sites (Supplementary Figure 4), we sought to confirm the accuracy of BCAM0996 T¹⁵⁹.
Examination of glycopeptides from BCAM0996 revealed that multiple Serine residues are
modified within the same peptide observed to be modified at T¹⁵⁹ (Supplementary Data 3 and 4).
The close proximity of this sole Threonine modification to multiple Serine modification events
further raised concerns of miss-localisation. Manual annotation of the assigned glycopeptide
supported the incorrect localisation of the glycosylation site, due to the incorrect assignment of
the glycan and a secondary modification within the peptide sequence, ultimately revealing S¹⁶⁷
to be the correct localisation site (Supplementary Figure 5). This finding suggested all localised
glycosylation sites within both *B. cenocepacia* strains were observed on Serine residues.
Examination of these 88 sites demonstrates that glycosylation favoured Alanine at the -1
position, yet this was not a strict requirement (Figure 2A).”

Specific points:

Line 46 “installation of glycans” – unusual terminology?

We have changed “installation of glycans” to “attachment of glycans” (Line 42)

52 “commonality” needs explaining to clarify

For clarity we have rewritten this section removing the use of the word commonality. This section
has been altered from

“Over the last decade, with the aid of mass spectrometry-based proteomics, a range of bacterial
glycosylation systems have been identified ^{1,2,10,11}. Although these studies have demonstrated
the commonality of bacterial glycosylation, the majority of follow up studies have focused on
understanding the biology of protein glycosylation glycans, including elucidating their
biosynthetic pathways ¹²⁻¹⁴ and defining their diversity ¹⁵⁻¹⁷.”

To

Line 46 to 50 “Over the last decade, with the aid of mass spectrometry-based proteomics, a range
of bacterial glycosylation systems have been identified with many of these now demonstrated to
be conserved across genera ^{1,2,10,11}. Within these systems follow up studies have mostly focused
on understanding the glycans used for protein glycosylation, defining their diversity ¹⁵⁻¹⁷ and
elucidating their biosynthetic pathways ¹²⁻¹⁴.”

62 “unique” consider word choice here and intended meaning

We thank the reviewer for highlighting this poor word choice. Although we used the word unique
to highlight the *Campylobacter* sequon is different from the typical N-linked sequon observed in
eukaryotic systems to improve the clarity of this sentence we have altered it to:

Line 57 to 59 “glycosylation events within the *Campylobacter* genus occurring within the
glycosylation sequon, D/E-X₁-N-X₂-S/T (where neither X₁ or X₂ can be proline)”

86 “predominately” predominantly?

This spelling issue has been corrected

107 “proteins” to protein

This spelling issue has been corrected

190 missing full stop

This punctuation issue has been corrected

229/231 “a” to “an”

This has been corrected

259 “deemed correct” consider rewording to clarify

We thank the reviewer for their suggestion and have changed “deemed correct” to “satisfying
the correctness of assignments guidelines outlined above” (Line 266).

319 “Iodacetamide” has initial capital – other chemicals don’t
The capital I in Iodoacetamide has been corrected to iodoacetamide (Line 324 and 327)
376 “contaminates” to contaminants?
This has been corrected (Line 380)
384 “statistically” to statistical?
This has been corrected (Line 388)
421 two full stops
This has been corrected
427/437 insert “respectively” as appropriate
This has been corrected
441 “predominately” again
This spelling issue has been corrected
446 How were these two proteins chosen?
As the *Burkholderia* O-linked glycans are modest in size (568 / 668 Da) we selected two small
proteins which would easily enable the confirmation of changes in mobility via western blotting.
In our hand’s proteins >50 kDa do not easily allow clear changes in mobility to be observed with
such small glycans.
450 Expression of the 2466 and 2345 proteins in the delta pglL strain results in increased gel
mobility rather than “reduced gel mobility” as stated. It would also be useful to mention that for
2466 both unglycosylated and glycosylated forms are observed in the PglL competent strain.
Some readers may need this help with interpretation here.
The reviewer raises an excellent point here and we have modified the discussion of the BCAL2466
and BCAL2345 results with the following sentence
Line 466 to 468 “Within the wildtype strains, differences in the occupancy of these proteins were
also noted, with no non-glycosylated protein observed for BCAL2345, contrasting BCAL2466
where both glycosylated and non-glycosylated forms were detected (Figure 1C)”

Reviewer #3

1. The last paragraph (line 628) of the discussion needs a little clarification, I am not sure I understand the final conclusions as written.

We appreciate the reviewer comment and have re-written the last paragraph to improve the clarity.

Line 660 to 673 “In summary, this work furthers our understanding of the breadth of the *B. cenocepacia* glycoproteome and the general features of glycosylation across members of the *Burkholderia* genus. The identification that the *B. cenocepacia* glycoproteome is far larger than initially thought, containing at least 98 proteins between strains, supports that glycosylation plays a multifaceted and pleiotropic role within *Burkholderia* species. The curation of a high-quality list of known glycoproteins and sites provides a unique resource to facilitate studies that work towards understanding the roles of glycosylation within glycoproteins. From a mechanistic and technical standpoint, the insights into the preference for glycosylation at Serine residues improves our ability to predict the specific sites likely to be modified in glycoproteins, as well as improves our ability to assign glycosylation within proteomic datasets. Finally, the demonstration that glycoproteins are highly conserved across *Burkholderia* species also provides a new opportunity to use comparative glycoproteomics to dissect the conserved roles of glycoproteins across this genus. Combined these insights will aid in future studies to understand why glycosylation events are common and have been maintained across *Burkholderia* species.”

REVIEWERS' COMMENTS:

Reviewer #1 (Remarks to the Author):

The authors have appropriately and thoroughly responded to my comments. I have no further concerns.

Ben Schulz

Reviewer #2 (Remarks to the Author):

I thank the authors for addressing my comments so thoroughly.

I have only one remaining issue. My comment regarding RNAPol mobility in Figure 1C that was also made by reviewer 1 (comment 6) was addressed through providing second repeat Western blot figures (Figure 1C). It was suggested that these "more ideal westerns" show no change in RpoA mobility when comparing wt and pglL knock out strains. I do not agree with this and there remain observable mobility shifts and so I cannot concur with the author's proposal that the new data demonstrate that "... this difference in mobility is an artefact.". This is not inherently problematic for the paper as a whole, as it might be considered a side issue but it has not been addressed as far as I can see.